# Effects of Citrulline or Watermelon Supplementation on Body Composition: A Systematic Review and Dose–Response Meta-Analysis

**DOI:** 10.3390/nu17193126

**Published:** 2025-09-30

**Authors:** Damoon Ashtary-Larky, Shooka Mohammadi, Seyed Amir Hossein Mousavi, Leila Hajizadeh, Darren G. Candow, Scott C. Forbes, Reza Afrisham, Vida Farrokhi, Jose Antonio, Katsuhiko Suzuki

**Affiliations:** 1Nutrition and Metabolic Diseases Research Center, Ahvaz Jundishapur University of Medical Sciences, Ahvaz 6135715794, Iran; 2Department of Social and Preventive Medicine, Faculty of Medicine, University of Malaya, Kuala Lumpur 50603, Malaysia; shooka.mohammadi@gmail.com; 3Student Research Committee, Ahvaz Jundishapur University of Medical Sciences, Ahvaz 6135715794, Iran; seyedamirmousavi1380@gmail.com; 4Department of Sport Physiology, Marvdasht Branch, Islamic Azad University, Marvdasht 7371113119, Iran; hajizadehleila266@gmail.com; 5Faculty of Kinesiology and Health Studies, University of Regina, Regina, SK S4S OA2, Canada; darren.candow@uregina.ca; 6Department of Physical Education Studies, Faculty of Education, Brandon University, Brandon, MB R7A 6A9, Canada; forbess@brandonu.ca; 7Department of Medical Laboratory Sciences, School of Allied Medical Sciences, Tehran University of Medical Sciences, Tehran 1417613151, Iran; rafrisham@sina.tums.ac.ir; 8Department of Hematology, Faculty of Allied Medicine, Tehran University of Medical Sciences, Tehran 1417613151, Iran; v-farrokhi@razi.tums.ac.ir; 9Department of Health and Human Performance, Nova Southeastern University, Davie, FL 32004, USA; jose.antonio@nova.edu; 10Faculty of Sport Sciences, Waseda University, Tokorozawa 359-1192, Japan

**Keywords:** L-Citrulline, body composition, anthropometric, watermelon, body fat percentage, fat-free mass

## Abstract

Background/Objectives: L-Citrulline (CIT) is a non-essential amino acid abundant in watermelon and commonly used as a dietary supplement to enhance exercise performance. Although its benefits for endurance and resistance training are well documented, its effects on body composition remain uncertain. This systematic review and dose–response meta-analysis aimed to assess the impact of CIT supplementation on anthropometric parameters. Methods: A comprehensive search of major databases identified relevant randomized controlled trials (RCTs) published until March 2025. A random-effects model was used to synthesize the data. Results: Twenty-one RCTs were included. Overall, CIT supplementation had no substantial effects on body mass index (BMI), body weight, fat mass (FM), waist circumference (WC), body fat percentage (BFP), and fat-free mass (FFM). Subgroup analyses revealed reductions in FM among participants over 40 years of age and in those administered more than 6 g/day of CIT. Interventions lasting 3 to 8 weeks were associated with a significant increase in FFM. Dose–response analyses suggested a non-linear association between CIT supplementation duration and changes in FM and FFM. Conclusions: CIT supplementation appears to have no overall effect on body composition. However, exploratory findings indicated potential benefits at higher doses or shorter durations. Rigorous trials controlling for dietary intake and training variables are needed to clarify its long-term effects.

## 1. Introduction

Obesity is a major global health concern associated with an increased risk of chronic conditions, such as type 2 diabetes, cardiovascular disease, and certain cancers [1,2,3,4]. The prevalence of obesity continues to rise, affecting both the general population and physically active individuals [5,6]. Multiple approaches, including dietary modifications, exercise strategies, and medical treatments, have been proposed to support weight loss [7,8]. In athletes, optimal body composition is critical for maximizing speed, endurance, and sport-specific performance [9]. Therefore, interventions that reduce fat mass (FM) while preserving or enhancing fat-free mass (FFM) are crucial for supporting health and optimizing athletic performance. Several nutritional and sports supplements have been proposed as potential strategies to improve body composition [10,11,12].

Citrulline (CIT) is a non-essential amino acid found in foods such as watermelon (*Citrullus lanatus*) and is produced endogenously as part of the urea cycle [13,14]. In its supplemental form, commonly known as L-CIT or CIT malate, it has gained popularity for its potential to enhance nitric oxide(NO) production [15], improve blood flow, reduce fatigue, and enhance exercise performance [16]. Several studies have reported that CIT supplementation can increase muscular strength, endurance, and overall training volume during resistance and endurance exercise sessions [17,18,19]. These performance benefits are often attributed to enhanced ammonia clearance, improved oxygen and nutrient delivery to the muscles, and reduced perception of exertion [16,20].

It is often hypothesized that supplements that improve exercise performance may indirectly lead to favorable changes in body composition by enabling higher training intensity, volume, or frequency, thus increasing energy expenditure. For example, creatine [21,22] and caffeine [23,24] have demonstrated consistent benefits for both performance and body composition in resistance-trained individuals. However, other popular ergogenic aids, such as beta-alanine [25], beetroot extract and dietary nitrates [26], and betaine [27], while effective for enhancing performance metrics, have shown limited or no significant impact on body composition, according to meta-analyses. Similarly, although some studies suggest that CIT may support improvements in body composition by enhancing training volume [28,29], some trials have reported no significant effects on FM or lean mass [30,31]. These inconsistencies highlight the need for a comprehensive evaluation.

Given the growing interest in CIT as a performance-enhancing supplement and the theoretical link between improved training capacity and body composition changes, a systematic review and meta-analysis are warranted to assess whether CIT supplementation meaningfully impacts body composition. Therefore, this systematic review and meta-analysis aimed to synthesize evidence from randomized controlled trials (RCTs) to determine whether the performance benefits attributed to CIT supplementation translate into measurable improvements in body composition.

## 2. Methods

This systematic review and meta-analysis was conducted in accordance with the Preferred Reporting Items for Systematic Reviews and Meta-Analyses (PRISMA) guidelines [32]. The study protocol was registered in the International Prospective Register of Systematic Reviews (PROSPERO; ID: CRD420251125585).

Two independent researchers conducted a comprehensive literature search to identify relevant RCTs published up to March 2025, without restrictions on publication date or language. The search included three major electronic databases: PubMed/MEDLINE, Scopus, and the Web of Science. The search strategy was developed based on the PICOS framework (Population, Intervention, Comparator, Outcomes, and Study Design) [33], as shown in Table 1. The following search terms were applied: (“citrulline” OR “citrulline supplementation” OR “CIT supplement” OR “ juice watermelon” OR “watermelon”) AND (“body composition” OR “obesity” OR “athletes” OR “body fat” OR “fat mass” OR “body mass” OR “body weight” OR “weight” OR “body mass index” OR “BMI” OR “fat free mass” OR “FFM” OR “FM” OR “lean mass” OR “lean body mass” OR “lean mass” OR “overweight” OR “obesity” OR “training” OR “exercise” OR “aerobic training” OR “resistance training” OR “high intensity interval training” OR “HIIT”).

### 2.1. Selection Criteria

All identified citations for this systematic review and meta-analysis were imported into the EndNote reference management software (version 21) for screening and organization. Two reviewers independently screened the studies according to the predefined inclusion criteria. Any disagreements were resolved through consultation with a third reviewer until consensus was reached.

This systematic review and meta-analysis included RCTs that examined the effects of CIT supplementation compared with a placebo or control group on body composition. Eligible trials were required to use a parallel or crossover design, involve an intervention period of at least two weeks, and report pre- and post-intervention data for both the CIT and placebo groups. The outcomes of interest included body mass index (BMI), body weight, FM, body fat percentage (BFP), waist circumference (WC), and FFM. Furthermore, studies were considered eligible if CIT was administered as a standalone supplement or if both groups received the same co-supplement, differing only by the addition of CIT.

RCTs were excluded if they did not include a control or placebo group, involved participants under 18 years of age or pregnant women, had a duration of less than two weeks, were observational or non-interventional in design, or did not provide sufficient baseline and follow-up data for specified outcomes.

### 2.2. Data Extraction

Two independent researchers extracted relevant data from the full texts of eligible studies. Any disagreements were resolved through discussions. The collected information encompassed key study characteristics, including study design, sample size, year of publication, location, trial duration, first author’s name, and administered dose of CIT supplements. Demographic data of the participants, including sex, average age, and mean BMI, were also collected. Additionally, outcome measures (including BMI, body weight, BFP, FM, WC, and FFM) were extracted at both baseline and the conclusion of the intervention period.

### 2.3. Statistical Analysis

All statistical analyses were performed using STATA software (version 17; Stata Corp LLC, College Station, TX, USA). The pooled effects of CIT supplementation on the selected outcomes were expressed as weighted mean differences (WMDs) with 95% confidence intervals (CIs), calculated from changes between baseline and post-intervention in both intervention and control groups. Results were presented as means with standard deviations (SDs), and effect sizes were reported as mean differences. When standard deviations for change values were not directly available, they were estimated using the following formula: SD change = √ [(SD^2^_baseline_ + SD^2^_post-intervention_) − (2 × R × SD_baseline_ × SD_post-intervention_)] [34], where R represents the assumed correlation between baseline and post-intervention measures [34]. A random-effects model was applied to compute pooled WMDs [34]. Statistical heterogeneity across studies was evaluated using Cochran’s Q test and the *I*^2^ statistic, with thresholds of <25% (low), 25–50% (moderate), 50–75% (high), and >75% (very high) [35].

Subgroup analyses were conducted to explore possible sources of heterogeneity, considering baseline outcome values, intervention duration (≤8 vs. >8 weeks), type of supplementation (CIT vs. watermelon), daily dosage (≤6 g/day vs. >6 g/day), baseline BMI category (normal, overweight, or obese), sex (male, female, or mixed), age (≤40 vs. >40 years), training status (trained vs. untrained), and type of training (high-intensity interval training [HIIT], resistance training [RT], and martial arts training [MA], and combined training [CT]).

Sensitivity analyses were conducted to assess the impact of individual trials on the overall findings. Publication bias was assessed using funnel plot inspection and further tested with Egger’s [36] and Begg’s [37] methods. A significance threshold of *p* < 0.05 was applied. Additionally, fractional polynomial models were used to examine potential non-linear associations between CIT supplementation dose or study duration and outcome changes. Meta-regression analyses were conducted to explore potential linear dose–response correlation between CIT supplementation dose or trial duration and outcome changes [38].

### 2.4. Quality Assessment

Two independent researchers assessed the quality of the trials using the Cochrane Risk of Bias 2 (RoB 2) tool [39]. This tool identified potential sources of bias, including reporting, performance, attrition, detection, and allocation biases. RoB for each domain was classified as low, unclear, or high [39].

### 2.5. GARDE

The certainty of the evidence was evaluated using the Grading of Recommendations Assessment, Development, and Evaluation (GRADE) framework, which is categorized into four levels: high, moderate, low, and very low [40].

## 3. Results

### 3.1. Study Selection

The database search initially identified 3817 records retrieved from PubMed (n = 1026), Web of Science (n = 861), and Scopus (n = 1930). After removing 813 duplicate entries, 3004 unique articles were screened. Following a review of the titles and abstracts, 2963 studies were excluded. Full-text evaluation was conducted for the remaining 41 articles to assess their eligibility. Of these, 20 records were excluded. Ultimately, 21 RCTs met the inclusion criteria and were included in the final meta-analysis [28,29,30,31,41,42,43,44,45,46,47,48,49,50,51,52,53,54,55,56,57]. The study selection process is illustrated in Figure 1.

### 3.2. Study Characteristics

Table 2 presents the characteristics of the studies included in the current meta-analysis. A total of 21 RCTs published between 2015 and 2023 were included in this meta-analysis [28,29,30,31,41,42,43,44,45,46,47,48,49,50,51,52,53,54,55,56,57]. The studies were conducted in various countries, including the USA [29,46,47,49,51,52,53], Iran [41,42,45,48,55,56,57], Canada [30,43,44,54], France [28,50], and Spain [31]. The study participants included patients with obesity [50,51,52,53,54], hypertension (HTN) [29,47], malnourished older patients [28], type 2 diabetes (T2DM) [42,56], and non-alcoholic fatty liver disease (NAFLD) [45]. Other participants included healthy postmenopausal women [46], older adults [30,43,44,48], elite taekwondo athletes [41,57], trained triathletes [31], elite wrestlers [55], and resistance-trained males [49].

The study designs were mostly parallel-group RCTs, with several employing double-blind and placebo-controlled methodologies. The intervention duration ranged from 3 to 12 weeks. The sample sizes varied from 16 to 81 participants. Both male and female participants were included, with some studies focusing on only one sex. The average baseline BMI of participants ranged widely across studies, from 19.7 to 35. CIT supplementation was provided either in the form of pure L-CIT [28,29,30,31,42,43,44,45,47,48,49,50,52,53,54,55,56] or watermelon juice (a natural source of CIT) [41,46,51,57], with daily doses ranging from 0.5 to 10 g. The control groups typically received placebo agents, such as maltodextrin, cellulose, or protein-matched comparators.

### 3.3. Effect of Supplementation with CIT on Body Weight and BMI

Sixteen effect sizes [29,30,31,41,42,45,46,49,51,52,53,54,55,56,57] from eligible trials were included to assess the impact of CIT supplementation on body weight. The pooled analysis revealed no statistically significant change in body weight following CIT supplementation compared with the control groups (WMD: 0.19 kg; 95% CI, −0.85, 1.24; *p* = 0.716) (Figure 2A and Table 3). Subgroup analyses showed no significant differences based on intervention duration (≤8 weeks, >8 weeks), CIT supplement type (CIT, WAT), supplement dose (≤6 g/day, >6 g/day), baseline BMI category (normal, overweight, or obese), sex (female, male, or both), age group (<40, ≥40 years), training status (yes, no), or training type (RT, HIIT, CT, MA) (Table 3).

The meta-analysis of 17 effect sizes [28,29,30,31,41,42,44,45,46,51,52,53,55,56,57] assessing the impact of CIT supplementation on BMI indicated no significant overall change (WMD: 0.01 kg/m^2^; 95% CI: −0.29, 0.31; *p* = 0.947) (Figure 2B and Table 3). Subgroup analyses also revealed no significant differences between the subgroups (Table 3).

### 3.4. Impacts of Supplementation with CIT on FM, BFP, and WC

A total of 11 effect sizes [28,30,31,43,44,49,50,54] were analyzed to determine the impact of CIT supplementation on FM. The overall effect was not statistically significant (WMD = −0.54 kg; 95% CI: −1.26, 0.18; *p* = 0.142) (Figure 2C and Table 3). Subgroup analysis revealed a significant reduction in FM among participants of both sexes, those who received CIT supplementation at doses greater than 6 g/day, and individuals over 40 years of age (Table 3).

Analysis of nine effect sizes [29,41,46,47,48,51,54,55,57] showed no significant overall effect of CIT on BFP (WMD = −0.24%; 95% CI: −0.76, 0.28; *p* = 0.367) (Figure 2D and Table 3). Data from six effect sizes [29,30,42,44,45,50] showed no significant reduction in WC with CIT supplementation (WMD = −0.54 cm; 95% CI: −2.02, 0.95; *p* = 0.480) (Figure 2E and Table 3). The subgroups did not yield any significant differences (Table 3).

### 3.5. Impact of Supplementation with CIT on FFM

A total of 11 effect sizes [28,30,31,43,44,49,50,55] were included in the meta-analysis that examined the effects of CIT supplementation on FFM. The pooled results indicated no significant overall effect (WMD = −0.22 kg; 95% CI: −1.69, 1.24; *p* = 0.763), with moderate heterogeneity across studies (*I*^2^ = 52.8%, *p* = 0.020) (Figure 2F, Table 3). Subgroup analysis based on trial duration showed a significant increase in FFM in studies lasting ≤8 weeks (Table 3).

### 3.6. Risk of Bias

The risk of bias was evaluated with the RoB 2 tool (Appendix A). Of the 21 RCTs included, 7 were classified as having a high RoB [41,47,51,52,53,55,57], while the other trials were rated as having a low RoB [28,29,30,31,42,43,44,45,46,48,49,50,54,56].

### 3.7. Publication Bias

Inspection of the funnel plots suggested some asymmetry for all assessed outcomes (BMI, body weight, FM, WC, BFP, and FFM) (Appendix A). Evidence of publication bias was found only for FM (Egger’s test, *p* = 0.041; Begg’s test, *p* = 0.043), whereas no bias was detected for BMI, body weight, or FFM. No formal tests for asymmetry were conducted for WC and BFP, which included fewer than 10 effect sizes; funnel plots are presented for completeness, and the findings should be interpreted descriptively rather than as conclusive evidence of publication bias.

### 3.8. Certainty of Evidence

The overall certainty of the evidence was evaluated using the GRADE framework. The quality of evidence for FM and FFM was rated as low due to concerns about imprecision, inconsistency, or potential publication bias. The certainty for body weight, BFP, and WC was downgraded to moderate owing to concerns related to imprecision or RoB. In contrast, BMI was rated as high certainty.

### 3.9. Linear and Non-Linear Dose–Response

The dose–response evaluation revealed a substantial non-linear correlation between changes in the duration of CIT supplementation and variations in FM (r = −0.38, *p* < 0.001; Appendix A) and FFM outcomes (r = 0.01, *p* = 0.017; Appendix A). No linear (Appendix A) or non-linear (Appendix A) correlations were observed between trial duration or CIT supplementation dose and changes in the other assessed outcomes.

### 3.10. Sensitivity Analysis

The leave-one-out sensitivity analysis indicated that the overall results were robust for most outcomes, as omitting any single study did not substantially alter the pooled effect sizes. The only exception was FM, where the removal of one study [31] led to a noticeable change in the effect size, suggesting that this study may have exerted a disproportionate influence on the overall FM outcome.

## 4. Discussion

This systematic review and meta-analysis found no significant effects of CIT supplementation on body weight, BMI, FM, BFP, WC, or FFM. These findings are consistent with previous meta-analyses reporting minimal or inconsistent effects of CIT supplementation on body composition outcomes [58].

The dose–response evaluation revealed a substantial non-linear correlation between changes in the duration of CIT supplementation and changes in FM and FFM outcomes. Subgroup analysis indicated a significant reduction in FM among participants of both sexes, those who received CIT supplementation at doses greater than 6 g/day, and individuals over 40 years of age. Furthermore, increases in FFM were observed in studies with intervention durations of ≤8 weeks; however, these changes may partly reflect transient shifts in body water or glycogen-associated water storage rather than true accretion of contractile muscle tissue [59,60]. These subgroup analyses should be viewed with caution, as the exclusion of other dietary supplements that may influence FFM or FM (i.e., protein, creatine, and caffeine) was not directly monitored.

CIT is mechanistically linked to NO production via the arginine NO pathway, which may enhance vasodilation, blood flow, and nutrient delivery during exercise [61,62]. It has been proposed that these physiological responses may improve muscle function and exercise performance by increasing oxygen availability, promoting ammonia clearance, and reducing fatigue [19,63,64]. It has been suggested that amino acid supplementation may influence body composition through more complex metabolic pathways. CIT or other amino acids may influence body composition by modulating insulin sensitivity and metabolic flexibility [65,66,67]. In addition to these effects, CIT may help improve sport-specific adaptations by increasing adenosine triphosphate (ATP) availability, promoting vasodilation, and enhancing the work capacity [68]. Moreover, CIT supplementation has been shown to improve physical performance, muscle strength, and walking speed [69,70,71]. However, the findings of this meta-analysis suggest that these performance benefits may not translate into measurable changes in body composition over time. This disconnect reflects prior findings on other ergogenic aids, such as beta-alanine [27] and beetroot extract/nitrates [26], which demonstrate ergogenic potential but have not consistently produced improvements in fat loss or lean mass accrual in meta-analyses. Thus, performance-enhancing effects alone are likely insufficient to produce substantial changes in body composition.

Improvements in body composition, particularly reductions in FM, increases in lean mass, or both, are primarily influenced by dietary intake and the duration and intensity of exercise training [72,73]. A supplement may influence body composition if it alters dietary intake, increases training volume, or is involved in anabolic pathways, such as muscle protein synthesis and fat oxidation. Although CIT does not appear to significantly affect appetite, dietary intake, or anabolic signaling pathways, it also seems insufficiently effective at enhancing training volume to a degree that would meaningfully increase FM and FFM. Regarding body weight, the ‘calories in, calories out’ model suggests that changes in body weight are predominantly determined by caloric intake rather than dietary supplementation [74,75]. A key limitation of the included studies is the lack of detailed reporting on participants’ dietary intake, caloric balance, and macronutrient distribution, which are critical determinants of body composition outcomes.

Interestingly, although the overall effects of CIT supplementation on body composition outcomes were not statistically significant in the main analyses, the non-linear dose–response analysis revealed two noteworthy associations: Specifically, a substantial reduction in FM was observed with increasing duration of supplementation, independent of dosage. This suggests that long-term CIT use may gradually induce beneficial changes in body fat, even in the absence of short-term effects. Additionally, a significant positive relationship was found between supplementation duration and FFM, indicating that prolonged CIT intake may help preserve or slightly enhance lean tissue. These non-linear trends, which were not detected in the linear regression models, highlight the potential time-dependent effects of CIT on body composition and suggest that metabolic adaptations to this supplement may require extended periods to manifest meaningfully.

It has been indicated that CIT supplementation may lead to modest yet significant enhancements in exercise performance and training volume, particularly during high-intensity and resistance-based activities [17,18,19]. While small increases in training intensity and volume may have limited short-term effects on body composition, these modest changes could accumulate over time and become more noticeable with ongoing training. For example, CIT-induced enhancements in exercise performance might lead to slightly higher energy expenditure during workouts [76], which, although minor, could contribute to gradual fat loss if maintained over time. Additionally, CIT has been shown to increase resistance training volume by helping individuals lift heavier weights and perform more repetitions [77]. Given that the primary determinant of hypertrophy is the number of sets performed near muscular failure, the capacity of CIT to enhance training volume may offer a modest yet significant additional stimulus for augmenting FFM over time. These cumulative physiological responses could help explain the positive link between the length of CIT supplementation and FFM observed in our non-linear dose–response analysis. This aligns with mechanistic insights from studies showing that metabolic adaptations to nutritional interventions often require sustained exposure beyond the usual durations of short-term trials. More long-term research is needed to explore the lasting effects of CIT supplementation on FM and FFM.

This study has several strengths, including a rigorous methodological approach, the application of GRADE to assess evidence certainty, and the evaluation of both linear and non-linear dose–response relationships. Most outcomes demonstrated moderate to high certainty of evidence. Moreover, a systematic search strategy was conducted without restrictions on publication date or language, ensuring comprehensive inclusion of relevant trials in the analysis. However, it also has certain limitations. The included trials were heterogeneous in terms of design, population, and intervention protocols. In addition, the overall quality of evidence for outcomes such as FM and FFM was rated as low. Furthermore, most studies did not control for dietary intake or apply standardized training protocols, limiting the ability to isolate the effect of CIT on body composition. Future trials should incorporate standardized nutritional assessment methods to better account for the confounding effects of nutrition on body composition outcomes. Techniques such as validated food frequency questionnaires, 24 h dietary recalls, and weighed food records can provide more accurate and consistent nutritional data. Moreover, several trials did not report performing sample size or power calculations, and their relatively small sample sizes may have limited their ability to detect meaningful outcomes. In addition, evidence of publication bias for FM outcomes and the disproportionate influence of a single study [31] further weakens the robustness of the findings. Furthermore, many studies did not provide information on test–retest reliability or the use of standardized protocols for measuring body composition and performance. This lack of methodological rigor may reduce the reliability and generalizability of the results. Another limitation is that some interventions used watermelon juice as a natural source of CIT. Although subgroup analyses were performed to distinguish between isolated CIT and watermelon sources, the presence of additional bioactive compounds and potential imprecision in estimating CIT content may have influenced the findings and should be interpreted with caution. Additionally, the sample sizes in several of the included trials [28,31,48,55] were relatively small, which may have limited their statistical power and increased variability in the results.

## 5. Conclusions

The present meta-analysis demonstrated that CIT supplementation does not significantly influence body composition parameters in untrained or recreationally active individuals. Although CIT has been shown to improve certain aspects of exercise performance, the evidence supporting its effects on meaningful fat loss or lean mass gain remains limited and inconclusive. Potential benefits in specific populations or with particular dosing protocols cannot be ruled out. Future trials should incorporate controlled dietary interventions and standardized training protocols to more accurately evaluate the long-term impact of citrulline on body composition.

## Figures and Tables

**Figure 1 nutrients-17-03126-f001:**
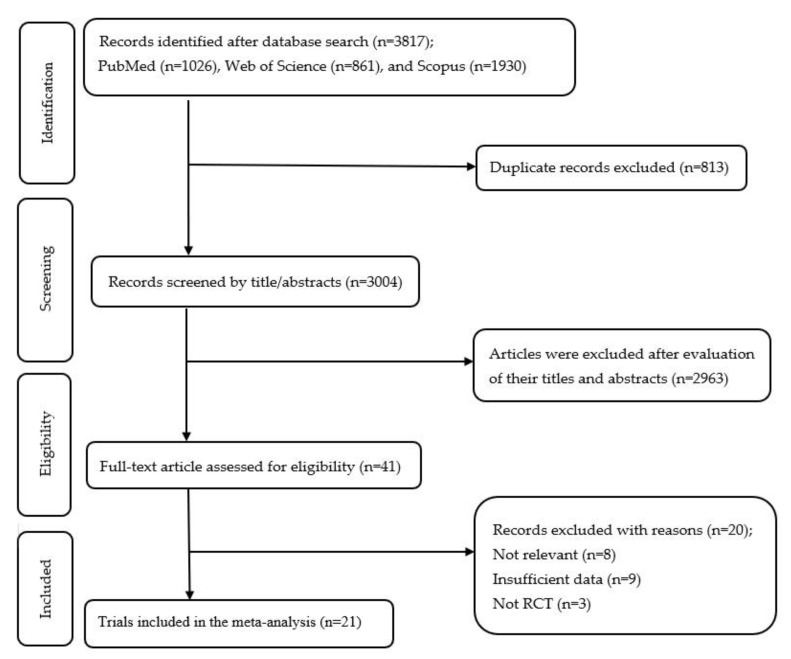
Flow diagram of study selection.

**Figure 2 nutrients-17-03126-f002:**
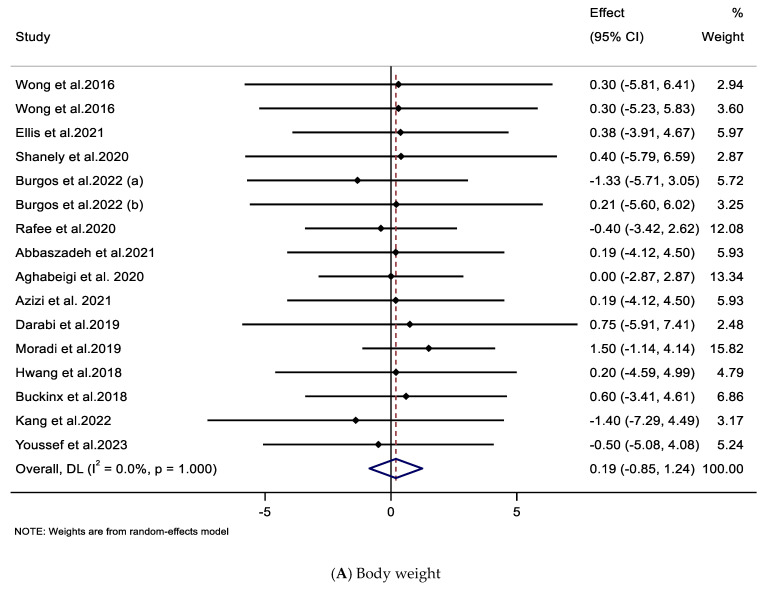
The forest plot illustrates the weighted mean difference and 95% confidence intervals (CIs) regarding the impacts of CIT supplementation on (**A**) body weight (Kg), (**B**) body mass index (kg/m^2^), (**C**) fat mass (kg), (**D**) body fat percentage (%), (**E**) waist circumference (cm), (**F**) fat-free mass (Kg). Diamond shapes represent pooled estimates from a random-effects analysis [28,29,30,31,41,42,43,44,45,46,47,48,49,50,51,52,53,54,55,56,57].

**Table 1 nutrients-17-03126-t001:** PICOS criteria were used for the inclusion of RCTs in the systematic review.

Component	Description	
Population	Adults (≥18 years), including healthy individuals, older adults, obese/overweight participants, trained athletes, and patients with metabolic disorders; excluding pregnant women	
Intervention	Citrulline supplementation as a standalone supplement (including pure L-citrulline or watermelon-derived citrulline)	
Comparison	Placebo or control group	
Outcome	Body composition parameters, including BMI, BFP, FM, body weight, WC, and FFM	
Study Design	RCTs, including both parallel and crossover designs, with an intervention duration of ≥2 weeks	

Abbreviations: BMI, body mass index; FM, fat mass; BFP, body fat percentage; WC, waist circumference; FFM, fat-free mass; RCTs, randomized controlled trials.

**Table 2 nutrients-17-03126-t002:** Characteristics of the included studies.

Studies	Country	Study Design	Participants	Sex	Sample Size		Trial Duration (Weeks)	Mean Age		Mean BMI		Intervention		
					IG	CG		IG	CG	IG	CG	Intervention Type	Supplement Dose (g/day)	CG
Wong et al. 2016 [53]	USA	P, R, PC	Obese postmenopausal women	♀ (23)	12	11	8	58 ± 4	58 ± 3	32.2 ± 2.4	32.9 ± 3.6	CIT	6	MD
Wong et al. 2016 [52]	USA	P, R, PC	Obese postmenopausal women	♀ (27)	13	14	8	58 ± 3	58 ± 4	33.8 ± 3.9	35 ± 3.4	CIT	6	MD
Ellis et al. 2021 [46]	USA	C, R, DB, PC	Healthy postmenopausal women	♀ (17)	17	17	4	60 ± 4	60 ± 4	25.0 ± 3.5	25.0 ± 3.5	WAT	720 mL/day (1.63 g)	Isocaloric placebo
Shanely et al. 2020 [51]	USA	P, R, CO	Obese postmenopausal women	♀ (45)	26	19	6	60 ± 5	60 ± 7	30.9 ± 4.6	30.3 ± 4.8	WAT	710 mL/day (1.61 g)	No intervention
Buckinx et al. 2020 [44]	Canada	P, R, DB, PC	Healthy older adults	♀/♂ (24/20)	23	21	12	68 ± 5	68 ± 3	26.1± 2.8	26.1 ± 2.2	CIT	10	MD
Bouillanne et al. 2019 (a) [28]	France	P, R, DB, PC	Malnourished older women	♀ (18)	8	10	3	89 ± 6	88 ± 4	19.9 ± 2.4	20.7 ± 3.7	CIT	10	Mixture of six NEAAs
Bouillanne et al. 2019 (b) [28]	France	P, R, DB, PC	Malnourished older patients	♀/♂ (18/6)	11	13	3	89 ± 6	88 ± 4	19.7 ± 2.5	21.6 ± 3.8	CIT	10	Mixture of six NEAAs
Marcangeli et al. 2022 [50]	France	P, R, DB, PC	Obese older adults	♀/♂ (43/38)	45	36	12	67 ± 5	68 ± 4	29.1 ± 4.3	29.3 ± 5.1	CIT	10	MD
Figueroa et al. 2015 [47]	USA	P, R, PC	Postmenopausal women with HTN	♀ (27)	13	14	8	58 ± 4	58 ± 4	33.8 ± 4	35 ± 3.4	CIT	6	MD
Rafee et al. 2020 [57]	Iran	P, R, SB, PC	Elite taekwondo athletes	♀ (25)	15	10	6	22 ± 2	21 ± 2	20.7 ± 1.9	20.9 ± 0.9	WAT	500 mL/day (1.17 g)	Placebo
Buckinx et al. 2019 (a) [43]	Canada	P, R, DB, PC	Sedentary obese older adults	♀/♂ (17/16)	19	14	12	68 ± 5	68 ± 4	30.4 ± 4	31.9 ± 6	CIT	10	MD + Protein
Buckinx et al. 2019 (b) [43]	Canada	P, R, DB, PC	Sedentary obese older adults	♀/♂ (23/17)	21	19	12	67 ± 5	68 ± 4	27.7 ± 5	27.6 ± 3.8	CIT	10	MD+ Protein
Burgos et al. 2022 (a) [31]	Spain	P, R, DB, PC	Trained male triathletes	♂ (16)	8	8	9	33 ± 7	34 ± 7	24.5 ± 2.5	24.0 ± 1.8	CIT	3	Cellulose
Burgos et al. 2022 (b) [31]	Spain	P, R, DB, PC	Trained male triathletes	♂ (16)	8	8	9	34 ± 8	33 ± 7	22.5 ± 1.6	23.2 ± 1.8	CIT	3	Nitrate-rich beetroot extract
Abbaszadeh et al. 2021 [56]	Iran	P, R, DB, PC	Patients with T2DM	♀/♂ (16/29)	23	22	8	48 ± 6	50 ± 5	29.7± 3.2	28.2 ± 2.1	CIT	3	Microcrystalline cellulose
Aghabeigi et al. 2020 [41]	Iran	P, R, SB, PC	Elite taekwondo players	♀ (25)	15	10	6	22 ± 2	22 ± 2	20.7 ± 1.9	20.9 ± 0.9	WAT	214 mL/day (0.5 g)	Placebo
Azizi et al. 2021 [42]	Iran	P, R, DB, PC	Patients with T2DM	♀/♂ (16/29)	23	22	8	48 ± 6	50 ± 5	29.7± 3.2	28.2 ± 2.1	CIT	3	Microcrystalline cellulose
Hosein Zade et al. 2021 [48]	Iran	P, R, DB, PC	Inactive elderly men	♂ (18)	9	9	8	64 ± 5	64 ± 5	28.6	28.6	CIT	6	Dextrose
Darabi et al. 2019 [45]	Iran	P, R, DB, PC	Patients with NAFLD	♀/♂ (21/23)	22	22	12	46 ± 12	45± 11	32.5 ± 6.6	33.5 ± 5.8	CIT	2	Starch
Moradi et al. 2019 [55]	Iran	P, R, SB, PC	Elite male wrestlers	♂ (19)	10	9	6	22 ± 2	22 ± 2	24.6 ± 0.5	23.8 ± 1.2	CIT	2	Cellulose
Hwang et al. 2018 [49]	USA	P, R, DB, PC	Resistance-trained males	♂ (50)	25	25	8	20 ± 2	20 ± 2	24.9	25.8	CIT	2	Cellulose
Buckinx et al. 2018 [30]	Canada	P, R, DB, PC	Dynapenic-obese older adults	♀/♂ (28/28)	26	30	12	66 ± 4	68 ± 4	30.5 ± 4.1	30.5 ± 4.9	CIT	10	MD
Kang et al. 2022 [29]	USA	P, R, DB, PC	Postmenopausal women with HTN	♀ (24)	13	11	8	62 ± 7	63 ± 3	29.6 ± 4	29.2 ± 5.6	CIT	10	MD
Youssef et al. 2023 [54]	Canada	P, R, DB, PC	Obese older adults	♀/♂ (59)	33	26	12	≥65	≥65	25	25.7	CIT	10	Placebo

Abbreviations: CIT, L-citrulline; WAT, watermelon; CO, controlled; IG, intervention group; SB, single-blinded; R, randomized; DB, double-blinded; NAFLD, non-alcoholic fatty liver disease; CG, control group; HTN, hypertension; T2DM, type 2 diabetes mellitus; MD, maltodextrin; ♀, female; ♂, male; NEAAs, non-essential amino acids; P, parallel; C, crossover; PC, placebo-controlled.

**Table 3 nutrients-17-03126-t003:** Subgroup analyses of the impacts of CIT supplementation on body composition and anthropometric parameters.

Sub-Groups	Number of Effect Sizes	WMD (95%CI)	*p*-Value	Heterogeneity		
				*p*-Value Heterogeneity	*I*^2^ (%)	*p*-Value Between Sub-Groups
Impacts of CIT supplementation on body weight (kg)						
Overall effect	16	0.19 (−0.85, 1.24)	0.716	1.000	0	
Trial duration (weeks)						
≤8	11	0.30 (−0.89, 1.50)	0.622	0.999	0	0.719
>8	5	−0.15 (−2.31, 2.00)	0.891	0.971	0	
Type of CIT						
CIT	12	0.31 (−0.97, 1.61)	0.630	0.999	0	0.750
WAT	4	−0.04 (−1.83, 1.75)	0.964	0.990	0	
Supplement dose (g/day)						
≤6	13	0.26 (−0.87, 1.40)	0.649	1.00	0	0.758
>6	3	−0.19 (−2.87, 2.49)	0.888	0.848	0	
Baseline BMI						
Normal	6	0.23 (−1.17, 1.64)	0.745	0.905	0	0.991
OW	2	0.28 (−2.75, 3.32)	0.854	0.951	0	
OB	8	0.09 (−1.72, 1.91)	0.918	1.000	0	
Sex						
Female	7	−0.08 (−1.67, 1.49)	0.913	0.999	0	0.864
Male	4	0.60 (−1.32, 2.52)	0.542	0.744	0	
Both	5	0.21 (−1.82, 2.25)	0.839	0.997	0	
Age						
≤40	6	0.23 (−1.17, 1.64)	0.745	0.905	0	0.934
>40	10	0.14 (−1.41, 1.70)	0.855	1.000	0	
Training						
Yes	10	0.15 (−1.07, 1.37)	0.810	0.992	0	0.887
No	6	0.32 (−1.73, 2.37)	0.758	1.000	0	
Training type						
RT	3	−0.20 (−3.29, 2.87)	0.895	0.897	0	0.911
HIIT	3	0.90 (−1.08, 2.88)	0.372	0.749	0	
CT	2	−0.77 (−4.27, 2.72)	0.666	0.679	0	
MA	2	−0.19 (−2.27, 1.89)	0.858	0.851	0	
Impacts of CIT supplementation on BMI (kg/m^2^)						
Overall effect	17	0.01 (−0.29.0.31)	0.947	0.993	0	
Trial duration (weeks)						
≤8	12	0.04 (−0.31, 0.39)	0.823	0.948	0	0.744
>8	5	−0.07 (−0.66, 0.51)	0.805	0.945	0	
Type of CIT						
CIT	13	0.12 (−0.22, 0.47)	0.501	0.993	0	0.226
WAT	4	−0.30 (−0.89, 0.28)	0.313	0.859	0	
Supplement dose (g/day)						
≤6	12	0.01 (−0.32, 0.35)	0.923	0.940	0	0.933
>6	5	−0.01 (−0.66, 0.63)	0.965	0.954	0	
Baseline BMI						
Normal	7	−0.04 (−0.45, 0.36)	0.829	0.545	0	0.893
OW	3	0.01 (−0.61, 0.64)	0.968	0.955	0	
OB	7	0.13 (−0.48, 0.76)	0.670	1.000	0	
Sex						
Female	8	−0.23 (−0.73, 0.26)	0.351	0.989	0	0.474
Male	3	0.22 (−0.37, 0.82)	0.462	0.333	9	
Both	6	0.07 (−0.43, 0.58)	0.772	0.997	0	
Age						
≤40	5	−0.03 (−0.53, 0.45)	0.880	0.326	13.8	0.785
>40	12	0.05 (−0.36, 0.46)	0.807	1.000	0	
Training						
Yes	9	−0.01 (−0.38, 0.36)	0.949	0.757	0	0.844
No	8	0.05 (−0.45, 0.55)	0.843	1.000	0	
Training type						
RT	2	0.09 (−1.43, 1.61)	0.905	0.853	0	0.491
HIIT	3	0.34 (−0.21, 0.90)	0.226	0.535	0	
CT	2	−0.19 (−1.04, 0.65)	0.648	0.582	0	
MA	2	−0.45 (−1.15, 0.23)	0.193	0.844	0	
Impacts of CIT supplementation on WC (cm)						
Overall effect	6	−0.54 (−2.02, 0.95)	0.480	0.974	0	
Trial duration (weeks)						
≤8	2	0.30 (−2.64, 3.24)	0.842	1.000	0	0.519
>8	4	−0.82 (−2.54, 0.90)	0.349	0.933	0	
Type of CIT						
CIT	6	−0.53 (−2.02, 0.95)	0.480	0.974	0	-
Supplement dose (g/day)						
≤6	2	−0.64 (−3.02, 1.73)	0.596	0.411	0	0.910
>6	4	−0.46 (−2.37, 1.43)	0.631	0.983	0	
Baseline BMI						
OW	2	−0.33 (−2.71, 2.05)	0.786	0.903	0	0.828
OB	4	−0.66 (−2.57, 1.23)	0.492	0.852	0	
Sex						
Female	1	0.30 (−6.39, 6.99)	0.930	-	-	0.802
Both	5	−0.58 (−2.10, 0.94)	0.456	0.940	0	
Age						
>40	6	−0.53 (−2.02, 0.95)	0.480	0.974	0	-
Training						
Yes	4	−0.46 (−2.37, 1.43)	0.631	0.983	0	0.910
No	2	−0.64 (−3.02, 1.73)	0.596	0.411	0	
Training type						
RT	1	0.30 (−6.39, 6.99)	0.930	-	-	0.967
HIIT	3	−0.53 (−2.52, 1.45)	0.598	0.948	0	
Impacts of CIT supplementation on FM (kg)						
Overall effect	11	−0.54 (−1.26, 0.18)	0.142	0.232	22.2	0.232
Trial duration (weeks)						
≤8	3	−2.08 (−5.23, 1.06)	0.194	0.041	68.7	0.264
>8	8	−0.25 (−0.90, 0.39)	0.440	0.691	0.0	
Type of CIT						
CIT	11	−0.54 (−1.26, 0.18)	0.142	0.232	62.3	-
Supplement dose (g/day)						
≤6	3	0.12 (−0.70, 0.95)	0.760	0.333	9.2	0.039
>6	8	−1.18 (−2.10, −0.25)	**0.012**	0.516	0	
Baseline BMI						
Normal	5	−0.64 (−2.11, 0.83)	0.392	0.022	65.1	0.972
OW	3	−0.88 (−2.16, 0.40)	0.178	0.878	0	
OB	3	−0.78 (−2.33, 0.76)	0.321	0.986	0	
Sex						
Female	1	−1.30 (−5.12, 2.52)	0.506	-	-	0.121
Male	3	0.12 (−0.70, 0.95)	0.760	0.333	9.2	
Both	7	−1.17 (−2.14, −0.20)	**0.017**	0.401	3.2	
Age						
≤40	3	0.12 (−0.70, 0.95)	0.760	0.333	9.2	0.039
>40	8	−1.18 (−2.10, −0.25)	**0.012**	0.516	0	
Training						
Yes	9	−0.24 (−0.86, 0.36)	0.425	0.784	0	0.126
No	2	−3.41 (−7.43, 0.59)	0.095	0.123	57.9	
Training type						
RT	1	−0.20 (−2.12, 1.72)	0.839	-	-	
HIIT	6	−0.84 (−1.83, 0.14)	0.095	0.998	0	0.265
CT	2	0.24 (−0.99, 1.49)	0.696	0.150	51.8	
Impacts of CIT supplementation on BFP (%)						
Overall effect	9	−0.24 (−0.76, 0.28)	0.367	0.992	0	
Trial duration (weeks)						
≤8	8	−0.21 (−0.74, 0.32)	0.441	0.989	0	0.627
>8	1	−0.80 (−3.12, 1.52)	0.500	-	-	
Type of CIT						
CIT	5	−0.51 (−1.35, 0.31)	0.221	0.964	0	0.398
WAT	4	−0.06 (−0.72, 0.60)	0.859	0.975	0	
Supplement dose (g/day)						
≤6	7	−0.18 (−0.72, 0.36)	0.515	0.988	0	0.451
>6	2	−0.91 (−2.74, 0.91)	0.328	0.876	0	
Baseline BMI						
Normal	3	−0.21 (−0.82, 0.40)	0.505	0.715	0	0.811
OW	2	0.15 (−1.58, 1.89)	0.861	0.866	0	
OB	4	−0.50 (−1.65, 0.63)	0.384	0.940	0	
Sex						
Female	6	−0.10 (−0.72, 0.52)	0.749	0.985	0	0.724
Male	2	−0.49 (−1.50, 0.52)	0.342	0.654	0	
Both	1	−0.80 (−3.12, 1.52)	0.500	-	-	
Age						
≤40	3	−0.21 (−0.82, 0.40)	0.505	0.715	0	0.867
>40	6	−0.30 (−1.26, 0.64)	0.529	0.976	0	
Training						
Yes	7	−0.25 (−0.80, 0.30)	0.373	0.972	0	0.896
No	2	−0.14 (−1.64, 1.35)	0.849	0.656	0	
Training type						
RT	2	−0.43 (−2.28, 1.42)	0.650	0.571	0	0.855
HIIT	2	−0.54 (−1.47, 0.38)	0.254	0.879	0	
MA	3	−0.03 (−0.78, 0.70)	0.917	0.988	0	
Impacts of CIT supplementation on FFM (kg)						
Overall effect	11	−0.22 (−1.69, 1.24)	0.763	0.020	52.8	
Trial duration (weeks)						
≤8	4	1.95 (0.43, 3.47)	**0.012**	0.863	0	0.004
>8	7	−1.35 (−2.99, 0.27)	0.103	0.119	40.9	
Type of CIT						
CIT	11	−0.22 (−1.68, 1.23)	0.763	0.020	52.8	
Supplement dose (g/day)						
≤6	4	−1.29 (−6.73, 4.14)	0.641	0.001	82.4	0.615
>6	7	0.12 (−0.99, 1.25)	0.823	0.669	0	
Baseline BMI						
Normal	6	−0.30 (−2.98, 2.92)	0.984	0.002	73.9	0.942
OW	3	−0.63 (−2.41, 1.14)	0.483	0.734	0	
OB	2	−0.41 (−2.52, 1.69)	0.702	0.819	0	
Sex						
Female	1	1.10 (−1.43, 3.63)	0.396	-	-	0.621
Male	4	−1.29 (−6.73, 4.14)	0.641	0.001	82.4	
Both	6	−0.10 (−1.36, 1.14)	0.865	0.646	0	
Age						
≤40	4	−1.29 (−6.73, 4.14)	0.641	0.001	82.4	0.615
>40	7	0.12 (−0.99, 1.25)	0.823	0.669	0	
Training						
Yes	9	−0.76 (−2.49, 0.96)	0.385	0.023	55.0	0.081
No	2	1.59 (−0.41, 3.59)	0.120	0.538	0	
Training type						
RT	1	0.52 (−15.69, 16.73)	0.950	-	-	0.325
HIIT	6	0.18 (−1.05, 1.42)	0.772	0.358	9.1	
CT	2	−3.81 (−9.66, 2.03)	0.201	0.056	72.7	

Abbreviations: CI, confidence interval; WMD, weighted mean differences; OW, overweight; OB, obesity; FFM, fat-free mass; BMI, body mass index; FM, fat mass; BFP, body fat percentage; WC, waist circumference; CIT: L-citrulline; WAT, watermelon; RT, resistance training; HIIT, high-intensity interval training; CT, combined training; MA: martial art training. Bold numbers indicate statistically significant differences (*p* < 0.05).

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
