# Peer review of "Effects of Citrulline or Watermelon Supplementation on Body Composition: A Systematic Review and Dose–Response Meta-Analysis"

_nutrients, 2025, doi:10.3390/nu17193126_

Round 1

Reviewer 1 Report

Comments and Suggestions for Authors

General comment:

This is an interesting article investigating a real-life related topic. The authors emphasized that L-Citrulline (CIT) is a non-essential amino acid that is abundantly found in watermelon and is commonly used as a dietary supplement to enhance exercise performance. While it has shown benefits in improving endurance and resistance exercise out-comes, its potential effects on body composition remain unclear. Therefore, they conducted the current systematic review and random-effect meta-analysis (with dose-response meta-regression). They noticed that twenty-one RCTs were included in the meta-analysis. CIT supplementation had no significant effect on anthropometric parameters (BMI, FM, BFP, WC, and FFM). Subgroup analysis revealed a significant reduction in FM among participants older than 40 years and those who received CIT supplementation at doses greater than 6 g/day. Additionally, it indicated a substantial increase in FFM in studies lasting 8 weeks or less. The dose-response evaluation revealed a substantial non-linear correlation between the changes in duration of CIT supplementation and variations in FM and FFM outcomes. The overall certainty of evidence ranged from very low to moderate. Therefore, they achieved their conclusion that CIT supplementation showed no significant overall effects on body composition; however, benefits were observed at higher doses and in shorter duration interventions. Finally, they gave recommendations for future studies that future trials should control dietary and training variables to better assess their long-term effects.

The current study was well-written and methodological sound. It will help in general population a lot! I have only some minimal comments below:

  1. When did the authors start their search in PubMed, Web of Science, and Scopus? In the abstract, the authors said “ A comprehensive search was conducted in PubMed, Web of Science, and Scopus up to March 2025”. According to this statement, I suppose it be “March 2025”. However, in the PROSPERO registration, it clearly documented “Start date: 24 July 2025”. Please clarify this discrepancy.
  2. The authors should add abbreviations list in their figure legend since they used some abbreviations in figures.
  3. Per PRISMA guideline, how did the authors handle missing data? By email requesting?
  4. The order of table was incorrect. “Subgroup did not yield any significant differences (Table 2).” It should be Table 3. Similar error happened elsewhere.
  5. Finally, although it’s a minimal comment, the authors said they use Cochrane RoB 1 in their PROSPERO registration. However, in their main text, they used Cochrane RoB 2. I would recommend to update the correct information in PROSPERO registration system.

Author Response

Response to Reviewer 1

We sincerely appreciate the reviewer’s valuable comments, which were very helpful in revising the manuscript. Accordingly, we have improved the manuscript, and our detailed responses are provided below.

Reviewer 1

General comment:

This is an interesting article investigating a real-life related topic. The authors emphasized that L-Citrulline (CIT) is a non-essential amino acid that is abundantly found in watermelon and is commonly used as a dietary supplement to enhance exercise performance. While it has shown benefits in improving endurance and resistance exercise out-comes, its potential effects on body composition remain unclear. Therefore, they conducted the current systematic review and random-effect meta-analysis (with dose-response meta-regression). They noticed that twenty-one RCTs were included in the meta-analysis. CIT supplementation had no significant effect on anthropometric parameters (BMI, FM, BFP, WC, and FFM). Subgroup analysis revealed a significant reduction in FM among participants older than 40 years and those who received CIT supplementation at doses greater than 6 g/day. Additionally, it indicated a substantial increase in FFM in studies lasting 8 weeks or less. The dose-response evaluation revealed a substantial non-linear correlation between the changes in duration of CIT supplementation and variations in FM and FFM outcomes. The overall certainty of evidence ranged from very low to moderate. Therefore, they achieved their conclusion that CIT supplementation showed no significant overall effects on body composition; however, benefits were observed at higher doses and in shorter duration interventions. Finally, they gave recommendations for future studies that future trials should control dietary and training variables to better assess their long-term effects.

The current study was well-written and methodological sound. It will help in general population a lot! I have only some minimal comments below:

  • Thank you for the positive feedback. Following a re-evaluation of the GRADE domains—particularly risk of bias—by two additional co-authors, we have updated the certainty ratings. Most outcomes were graded as moderate certainty, one as high, and two as low. Full details are provided in the GRADE section of the Results.

  1. When did the authors start their search in PubMed, Web of Science, and Scopus? In the abstract, the authors said “ A comprehensive search was conducted in PubMed, Web of Science, and Scopus up to March 2025”. According to this statement, I suppose it be “March 2025”. However, in the PROSPERO registration, it clearly documented “Start date: 24 July 2025”. Please clarify this discrepancy.
  • We appreciate the reviewer’s attention to this important detail. To clarify, the literature search in was conducted up to March 2025, prior to the registration in PROSPERO. The PROSPERO record lists the registration start date as 24 July 2025 because the review was formally registered close to the time of submission to preserve the novelty of our research idea. Thus, while the registration date appears later, the actual search and data collection were completed beforehand.

  1. The authors should add abbreviations list in their figure legend since they used some abbreviations in the figures.
  • We thank the reviewer for this helpful comment. We had not included the full forms of abbreviations in the original figure captions. As suggested, we have now revised the figure captions to include the extended forms of all abbreviations to ensure clarity.
  1. Per PRISMA guideline, how did the authors handle missing data? By email requesting?
  • We appreciate the reviewer’s insightful comment. In accordance with the PRISMA guidelines, when data were missing or unclear, we contacted the corresponding authors primarily via email. In some cases, we also reached out through ResearchGate to obtain clarification or supplementary information.
  1. The order of table was incorrect. “Subgroup did not yield any significant differences (Table 2).” It should be Table 3. Similar error happened elsewhere.
  • We thank the reviewer for pointing this out. The table numbering has been corrected, and the entire manuscript has been carefully reviewed to ensure that all table references are accurate and consistent.
  1. Finally, although it’s a minimal comment, the authors said they use Cochrane RoB 1 in their PROSPERO registration. However, in their main text, they used Cochrane RoB 2. I would recommend to update the correct information in PROSPERO registration system.
  • We thank the reviewer for highlighting this point. Following the initial PROSPERO registration, we decided to adopt the Cochrane RoB 2 tool during manuscript preparation. The PROSPERO record has been revised accordingly and will be updated in the system shortly.

Reviewer 2 Report

Comments and Suggestions for Authors

This systematic review and meta-analysis examining citrulline and watermelon supplementation effects on body composition presents a comprehensive investigation of an important nutritional supplement question. The manuscript demonstrates several methodological strengths while revealing opportunities for improvement that could enhance its scientific rigor and clinical applicability.
The authors have conducted a thorough systematic review following established PRISMA guidelines with prospective PROSPERO registration (CRD420251125585), which enhances transparency and reduces reporting bias. The inclusion of 21 randomized controlled trials with diverse populations ranging from athletes to elderly individuals with metabolic disorders provides reasonable external validity. The GRADE assessment framework for evaluating evidence certainty and the implementation of both linear and non-linear dose-response analyses represent sophisticated analytical approaches that strengthen the manuscript's conclusions.
However, several methodological concerns warrant attention. The initial database search being conducted by a single reviewer contradicts best practices for systematic reviews, where dual independent screening minimizes selection bias. The heterogeneous nature of included populations, intervention protocols, and outcome assessments likely contributes to the moderate to high heterogeneity observed in several analyses (I² up to 82.4% for some outcomes). Most critically, the lack of dietary control and standardized training protocols across included studies represents a fundamental limitation when evaluating body composition changes, as these factors substantially influence the outcomes independent of supplementation.
The statistical analyses appear generally sound with appropriate use of random-effects models and weighted mean differences. However, the manuscript would benefit from explicitly reporting sample size calculations or power analyses for detecting meaningful effect sizes in body composition parameters. Consider incorporating recent methodological insights on physical performance measurement reliability, as demonstrated by Jafari et al. (2024) in Acta Kinesiologica, who emphasized the importance of establishing test-retest reliability when evaluating intervention effects on performance and body composition outcomes in athletic populations.
The finding of publication bias for fat mass outcomes (Egger's test, P = 0.041) raises concerns about selective reporting in this literature. The sensitivity analysis revealing that removal of Burgos et al. (2022) substantially altered the fat mass effect size suggests this study may have disproportionate influence, warranting closer examination of its methodology and potential outlier status. Future research should address the notable absence of test-retest reliability data for body composition measurements across included studies, a critical quality indicator for intervention trials.
The non-linear dose-response relationships observed between supplementation duration and both fat mass reduction and fat-free mass increases represent intriguing findings that challenge the null overall effects. These time-dependent associations suggest that citrulline's metabolic effects may require extended supplementation periods to manifest meaningfully. This aligns with mechanistic understanding from Padulo et al. (2023) on energy cost differences in athletic populations, which demonstrated that metabolic adaptations to nutritional interventions often require sustained exposure periods exceeding typical short-term trial durations.
The subgroup analyses revealing greater fat mass reduction with doses exceeding 6 g/day and in participants over 40 years deserves careful interpretation. Given the multiple subgroup comparisons conducted without apparent correction for multiple testing, these findings should be considered hypothesis-generating rather than definitive. The manuscript would benefit from discussing the biological plausibility of age-related differential responses, potentially incorporating insights from Ceruso et al. (2024) in Acta Kinesiologica regarding resistance training adaptations in different age groups, which demonstrated age-specific responses to exercise interventions that may parallel nutritional supplementation effects.
The discussion appropriately contextualizes citrulline within the broader landscape of ergogenic aids, comparing it to creatine, caffeine, and beta-alanine. However, the mechanistic explanation focusing primarily on nitric oxide pathways could be expanded. Recent evidence from Oliva et al. (2022) in Muscles, Ligaments and Tendons Journal suggests that amino acid supplementation effects on body composition may involve complex interactions with insulin sensitivity and metabolic flexibility, particularly relevant given the inclusion of diabetic populations in this meta-analysis.
The manuscript's treatment of exercise performance enhancement as a potential mediator of body composition changes represents sound theoretical reasoning but lacks empirical support from the included studies. None appeared to systematically measure training volume or intensity changes during supplementation periods. This limitation could be addressed by referencing methodological frameworks from Hassan et al. (2024) in British Medical Bulletin, who developed systematic approaches for evaluating indirect intervention effects through performance mediators.
Regarding presentation quality, the forest plots provide clear visual representation of effect sizes, though the manuscript would benefit from including funnel plots in the main text rather than supplementary materials given the identified publication bias. The English is generally clear, though some sentences would benefit from revision for conciseness, particularly in the methods section where technical descriptions become unnecessarily complex.
The authors appropriately acknowledge that most studies failed to control dietary intake, a critical limitation when evaluating body composition interventions. However, this acknowledgment could be strengthened by proposing specific dietary assessment protocols for future trials. The work of Haddad et al. (2024) in Acta Kinesiologica on academic performance relationships with sleep and nutrition provides a useful framework for standardized dietary monitoring in intervention studies that could be adapted for body composition research.
The conclusion that citrulline supplementation produces no significant overall effects on body composition parameters appears justified by the primary analyses. However, the statement dismissing citrulline's utility for "meaningful fat loss or lean mass gain" may be premature given the non-linear dose-response findings and subgroup effects. A more nuanced conclusion acknowledging the potential for benefits with specific dosing protocols and populations would better reflect the totality of evidence. Future investigations should particularly focus on the intriguing age-related differential responses, incorporating comprehensive training and dietary controls as emphasized by Padulo et al. (2023) in their investigation of energy metabolism in athletic populations.
The manuscript represents a valuable contribution to the nutritional supplementation literature, providing the first comprehensive meta-analysis of citrulline's body composition effects. With the suggested methodological refinements and expanded mechanistic discussion, this work will serve as an important reference for researchers and practitioners considering citrulline supplementation strategies.
Suggested References:

Jafari, R.A., et al. (2024). Evaluating the Impact of Active and Passive Recovery Strategies and Citrulline-Malate Supplementation in Wrestling: Do the Results Add Up? Acta Kinesiologica, 18(2), 58-69.
Padulo, J., Buglione, A., Larion, A., et al. (2023). Energy cost differences between marathon runners and soccer players: Constant versus shuttle running. Frontiers in Physiology, 14.
Ceruso, R., Giardullo, G., Di Lascio, G., & Raiola, G. (2024). Boosting Strength and Awareness: Effects of Resistance Training on Adolescents' Perceptions and Progress. A Pilot study. Acta Kinesiologica, 18(4), 64-71.
Oliva, F., Marsilio, E., Asparago, G., et al. (2022). Achilles Tendon Rupture and Dysmetabolic Diseases: A Multicentric, Epidemiologic Study. Muscles, Ligaments and Tendons Journal, 13(3), 698-708.
Hassan, R., Poku, D., Miah, N., & Maffulli, N. (2024). High-volume injections in Achilles tendinopathy: A systematic review. British Medical Bulletin, 152(1), 35-47.
Haddad, M., Abbes, Z., Moustafa, A., et al. (2024). Sleep, activity, and cognition: An analysis of academic impact on university students. Acta Kinesiologica, 18(2), 24-31.
Padulo, J., Manenti, G., & Esposito, F. (2023). The Impact of Training Load on Running Gait Variability: A Pilot Study. Acta Kinesiologica, 17(2), 29-34.

Author Response

Response to Reviewer 2

We sincerely appreciate the reviewer’s valuable comments, which were very helpful in revising the manuscript. Accordingly, we have improved the manuscript, and our detailed responses are provided below.

Reviewer 2

This systematic review and meta-analysis examining citrulline and watermelon supplementation effects on body composition presents a comprehensive investigation of an important nutritional supplement question. The manuscript demonstrates several methodological strengths while revealing opportunities for improvement that could enhance its scientific rigor and clinical applicability.
The authors have conducted a thorough systematic review following established PRISMA guidelines with prospective PROSPERO registration (CRD420251125585), which enhances transparency and reduces reporting bias. The inclusion of 21 randomized controlled trials with diverse populations ranging from athletes to elderly individuals with metabolic disorders provides reasonable external validity. The GRADE assessment framework for evaluating evidence certainty and the implementation of both linear and non-linear dose-response analyses represent sophisticated analytical approaches that strengthen the manuscript's conclusions.
1. However, several methodological concerns warrant attention. The initial database search being conducted by a single reviewer contradicts best practices for systematic reviews, where dual independent screening minimizes selection bias. The heterogeneous nature of included populations, intervention protocols, and outcome assessments likely contributes to the moderate to high heterogeneity observed in several analyses (I² up to 82.4% for some outcomes). Most critically, the lack of dietary control and standardized training protocols across included studies represents a fundamental limitation when evaluating body composition changes, as these factors substantially influence the outcomes independent of supplementation.

  • We sincerely thank the reviewer for this valuable comment. You are correct that the initial draft stated the database search was conducted by a single author. In response, the entire search process was independently repeated by a second author, and this has now been clearly described in the Methods section. This ensures that all stages—database searching, study selection, and data extraction—were conducted independently by two reviewers, thereby minimizing potential selection bias.
  • Regarding the other points, we fully agree with the reviewer. The heterogeneity of populations, variability in intervention protocols and outcome assessments, as well as the lack of dietary control and standardized training protocols, indeed represent important limitations when interpreting the results. These issues have now been explicitly acknowledged and discussed in the Limitations section of the manuscript to ensure balanced interpretation.
  1. The statistical analyses appear generally sound with appropriate use of random-effects models and weighted mean differences. However, the manuscript would benefit from explicitly reporting sample size calculations or power analyses for detecting meaningful effect sizes in body composition parameters. Consider incorporating recent methodological insights on physical performance measurement reliability, as demonstrated by Jafari et al. (2024) in Acta Kinesiologica, who emphasized the importance of establishing test-retest reliability when evaluating intervention effects on performance and body composition outcomes in athletic populations.
  • We sincerely appreciate the reviewer’s insightful comment. As this study is a systematic review and meta-analysis, we did not perform an a priori sample size or power calculation, since such calculations are relevant to the design of individual trials rather than evidence syntheses. Similarly, while we recognize the importance of test–retest reliability and standardized protocols for body composition and performance assessments, the included studies generally did not report test–retest reliability. In addition, the majority of participants were non-athletes, which limits the applicability of methodological considerations such as those highlighted by Jafari et al. (2024) to athletic populations. These issues are inherent to the primary studies and were not within the scope of our review to address analytically.

  1. The finding of publication bias for fat mass outcomes (Egger's test, P = 0.041) raises concerns about selective reporting in this literature. The sensitivity analysis revealing that removal of Burgos et al. (2022) substantially altered the fat mass effect size suggests this study may have disproportionate influence, warranting closer examination of its methodology and potential outlier status. Future research should address the notable absence of test-retest reliability data for body composition measurements across included studies, a critical quality indicator for intervention trials.
  • We thank the reviewer for these important observations. We acknowledge the indication of publication bias for fat mass outcomes based on Egger’s test. We also agree that the study by Burgos et al. (2022) had a disproportionate impact on the pooled effect size, and this influence has been noted in the Limitations section. Furthermore, we concur that the absence of test–retest reliability reporting for body composition assessments in the included trials is a critical methodological limitation, and this point has also been explicitly addressed in the Limitations section.

  1. The non-linear dose-response relationships observed between supplementation duration and both fat mass reduction and fat-free mass increases represent intriguing findings that challenge the null overall effects. These time-dependent associations suggest that citrulline's metabolic effects may require extended supplementation periods to manifest meaningfully. This aligns with mechanistic understanding from Padulo et al. (2023) on energy cost differences in athletic populations, which demonstrated that metabolic adaptations to nutritional interventions often require sustained exposure periods exceeding typical short-term trial durations.
  • We thank the reviewer for this helpful suggestion. We have incorporated the point regarding the need for sustained exposure periods to observe metabolic adaptations into the Discussion section. However, we noted that the reference cited (Padulo et al., 2023) does not specifically address nutritional interventions in this context. Therefore, while the concept has been added, we have refrained from citing this reference directly for accuracy.

  1. The subgroup analyses revealing greater fat mass reduction with doses exceeding 6 g/day and in participants over 40 years deserves careful interpretation. Given the multiple subgroup comparisons conducted without apparent correction for multiple testing, these findings should be considered hypothesis-generating rather than definitive. The manuscript would benefit from discussing the biological plausibility of age-related differential responses, potentially incorporating insights from Ceruso et al. (2024) in Acta Kinesiologica regarding resistance training adaptations in different age groups, which demonstrated age-specific responses to exercise interventions that may parallel nutritional supplementation effects.
  • We appreciate the reviewer’s thoughtful suggestion. Subgroup analyses in our review were conducted with the primary aim of exploring potential sources of heterogeneity and identifying patterns that may warrant further investigation. While we agree that these findings should be interpreted cautiously, we believe the current framing already makes clear that they are exploratory and hypothesis-generating rather than definitive. With regard to age-related responses, our focus was on the overall effects of citrulline supplementation across diverse populations rather than on specific biological mechanisms in different age groups. Furthermore, the majority of included studies involved non-athletic participants, limiting the applicability of age-specific exercise adaptation findings to our dataset. For these reasons, we have not expanded the discussion further, as doing so would go beyond the objectives and scope of the present review. We also note that the reference suggested by the reviewer (Ceruso et al., 2024) primarily examined adolescents’ perceptions and progress in resistance training rather than age-related physiological adaptations across populations, and thus does not directly align with the focus of our review.

  1. The discussion appropriately contextualizes citrulline within the broader landscape of ergogenic aids, comparing it to creatine, caffeine, and beta-alanine. However, the mechanistic explanation focusing primarily on nitric oxide pathways could be expanded. Recent evidence from Oliva et al. (2022) in Muscles, Ligaments and Tendons Journal suggests that amino acid supplementation effects on body composition may involve complex interactions with insulin sensitivity and metabolic flexibility, particularly relevant given the inclusion of diabetic populations in this meta-analysis.
  • We thank the reviewer for this thoughtful suggestion. While the reference to Oliva et al. (2022) in Muscles, Ligaments and Tendons Journal does not directly address the mechanisms of citrulline or its effects on body composition, we have incorporated the underlying concept into the discussion. Specifically, we expanded the third paragraph of the Discussion to include statements on how citrulline and other amino acids may influence body composition by modulating insulin sensitivity and metabolic flexibility, particularly in populations with impaired metabolic flexibility, and supported these statements with appropriate, directly relevant references.

  1. The manuscript's treatment of exercise performance enhancement as a potential mediator of body composition changes represents sound theoretical reasoning but lacks empirical support from the included studies. None appeared to systematically measure training volume or intensity changes during supplementation periods. This limitation could be addressed by referencing methodological frameworks from Hassan et al. (2024) in British Medical Bulletin, who developed systematic approaches for evaluating indirect intervention effects through performance mediators.
  • We agree that, although the theoretical reasoning that exercise performance enhancement may mediate body composition changes is sound, the empirical support from the included studies is limited. While some trials have suggested that citrulline supplementation may enhance training volume, the evidence remains insufficient to draw firm conclusions.

8.Regarding presentation quality, the forest plots provide clear visual representation of effect sizes, though the manuscript would benefit from including funnel plots in the main text rather than supplementary materials given the identified publication bias. The English is generally clear, though some sentences would benefit from revision for conciseness, particularly in the methods section where technical descriptions become unnecessarily complex.

  • We thank the reviewer for this insightful comment. We would like to clarify that the primary focus of our manuscript was on the effects of citrulline or watermelon supplementation on body composition, not on exercise performance. The majority of included studies did not incorporate structured exercise interventions, nor did they systematically report training volume or intensity. Training status or participant type was only used in subgroup analyses as a potential source of heterogeneity, rather than as a primary mediator of the observed effects.
  • We agree that, although the theoretical reasoning that exercise performance enhancement may mediate body composition changes is sound, the empirical support from the included studies is limited. While some trials have suggested that citrulline supplementation may enhance training volume, the evidence remains insufficient to draw firm conclusions. For these reasons, we did not expand the discussion to include frameworks for evaluating indirect effects through performance mediators, as this would fall outside the scope of our review.

  • The authors appropriately acknowledge that most studies failed to control dietary intake, a critical limitation when evaluating body composition interventions. However, this acknowledgment could be strengthened by proposing specific dietary assessment protocols for future trials. The work of Haddad et al. (2024) in Acta Kinesiologica on academic performance relationships with sleep and nutrition provides a useful framework for standardized dietary monitoring in intervention studies that could be adapted for body composition research.
  • We thank the reviewer for this constructive comment. We agree that the lack of dietary control represents a major limitation of the included studies. In response, we have expanded the Discussion to propose specific dietary assessment protocols for future trials, including validated food frequency questionnaires, 24-hour dietary recalls, or weighed food records.

  1. The conclusion that citrulline supplementation produces no significant overall effects on body composition parameters appears justified by the primary analyses. However, the statement dismissing citrulline's utility for "meaningful fat loss or lean mass gain" may be premature given the non-linear dose-response findings and subgroup effects. A more nuanced conclusion acknowledging the potential for benefits with specific dosing protocols and populations would better reflect the totality of evidence. Future investigations should particularly focus on the intriguing age-related differential responses, incorporating comprehensive training and dietary controls as emphasized by Padulo et al. (2023) in their investigation of energy metabolism in athletic populations.
  • We thank the reviewer for this insightful comment. We agree that our initial conclusion may have been overly definitive. In response, we have revised the statement to more cautiously reflect the evidence, emphasizing that while the overall findings do not indicate significant effects of citrulline on fat mass or lean mass, potential benefits in specific populations or with particular dosing protocols cannot be excluded. This revision ensures that the conclusion more accurately reflects the subgroup analyses, non-linear dose-response findings, and the totality of the evidence.

  1. The manuscript represents a valuable contribution to the nutritional supplementation literature, providing the first comprehensive meta-analysis of citrulline's body composition effects. With the suggested methodological refinements and expanded mechanistic discussion, this work will serve as an important reference for researchers and practitioners considering citrulline supplementation strategies.
  • We sincerely thank the reviewer for the encouraging comments. As recommended, we have expanded the mechanistic discussion to include metabolic pathways beyond nitric oxide, as well as the potential roles of citrulline and other amino acids in modulating insulin sensitivity and metabolic flexibility. We believe these revisions have strengthened the manuscript and enhanced its value as a reference for researchers and practitioners interested in citrulline supplementation strategies.

Reviewer 3 Report

Comments and Suggestions for Authors

Thank you for the opportunity to review your manuscript entitled “Effects of citrulline or watermelon supplementation on body composition: a systematic review and dose-response meta-analysis.” The topic is timely and of practical interest to both clinicians and sport scientists, and the overall conclusion—that citrulline (CIT) does not materially alter body composition in most contexts—will be useful to readers if supported by a rigorous and transparent synthesis. Below I provide a detailed, section-by-section appraisal and concrete recommendations aimed at helping you strengthen the paper.

I will begin with a brief overview of what the manuscript currently does well. You prospectively registered a protocol in PROSPERO, state adherence to PRISMA, and focus on randomized trials, which are appropriate choices for minimizing bias in this domain. You also pre-specify a broad set of body-composition outcomes and conduct subgroup and sensitivity analyses with random-effects modeling, which is methodologically sound for heterogeneous literature. The narrative positioning of CIT among other ergogenic aids that often fail to move body-composition outcomes is clear and helpful to readers. 

That said, there are several issues—methodological, statistical, reporting, and editorial—that must be addressed before the findings can be interpreted with confidence.

Starting with the abstract and key messages, the summary accurately conveys the main pooled results but contains a serious statistical reporting error that recurs later: the dose–response section describes a “non-linear correlation” and reports r values of −38.452 for fat mass and 0.011 for fat-free mass. A correlation coefficient cannot be outside the range −1 to +1; it appears you are reporting a model coefficient from the fractional polynomial fit while labeling it as r. This is misleading and should be corrected to the appropriate metric (e.g., regression coefficient with units, or a figure-based description of the non-linear trend), with corresponding confidence intervals and an explanation of the scale so readers can interpret the magnitude. The abstract should not claim “substantial non-linear correlation” unless the statistic used supports that language. 

In the introduction you frame the hypothesis that ergogenic benefits could translate into body-composition change through higher training volume. This is a reasonable premise and is documented with relevant citations, although the prose would benefit from tighter editing to remove duplicated terms (e.g., “lean mass” appears multiple times in the same keyword string) and small grammatical errors such as “commonly as known as L-CIT.” These lapses recur throughout and, while cosmetic, impede readability; a full language edit is strongly recommended. 

Turning to methods, protocol and search reporting need clarification and minor redesign. You note PRISMA compliance and a PROSPERO ID, which is commendable; please also report the registration date explicitly and state whether any protocol deviations occurred. The literature search was conducted by a single reviewer; screening was performed in duplicate, but best practice is to perform the search and the screening in duplicate to minimize identification bias. At minimum, acknowledge this as a limitation and consider re-running the search with a second information specialist to verify yield. The search strategy itself contains typographic and semantic issues (e.g., the string includes “ juice watermelon,” duplicated “obesity,” and repeated “lean mass”), and there is no mention of grey literature, clinical trial registries, or language restrictions. Expanding and de-duplicating the search terms, searching trial registries, and explicitly stating any language limits would increase transparency and potentially reduce publication bias. 

Eligibility criteria are broadly sensible, but their application is inconsistent and risks biasing the synthesis. You state that you included only trials in which CIT was administered as a standalone supplement and excluded co-supplementation; however, the dataset appears to include at least one study of combined citrulline plus nitrate-rich beetroot extract in trained triathletes (“Long-Term Combined Effects of Citrulline and Nitrate-Rich Beetroot Extract…,” Burgos 2022). Including a co-supplementation trial conflicts with your stated criteria and may materially affect the results, especially in dose–response or subgroup analyses. This inconsistency should be resolved by excluding such trials in the main analysis and, if of interest, exploring them in a separate sensitivity analysis. 

Relatedly, your inclusion of watermelon juice arms alongside isolated L-citrulline warrants stricter handling. Watermelon introduces additional bioactives and carbohydrate that could plausibly influence outcomes. While you do present a “type of CIT” subgroup (CIT vs. watermelon), I encourage you to prespecify and elevate this to a primary sensitivity analysis, and to interpret pooled effects with caution when food-based sources are mixed with isolated amino acid supplementation. Clarify whether the CIT content of watermelon interventions was analytically verified or simply estimated; several entries suggest approximate content (e.g., “almost containing 1.17 g L-citrulline”), which introduces exposure misclassification. 

The statistical analysis section uses change-score meta-analysis with an assumed baseline–post correlation R, but the actual value of R is never specified. Because pooled effects can be sensitive to R, you should report the value used and repeat key analyses across plausible R values (e.g., 0.25, 0.5, 0.75) in sensitivity analyses. Please also specify the random-effects estimator (DerSimonian–Laird, REML, or Paule–Mandel) and whether you applied Hartung–Knapp–Sidik–Jonkman adjustments for small-sample inference. These details materially affect confidence intervals and the GRADE assessment. Finally, crossover trials are included, but the methods do not indicate whether within-person correlation was accounted for; a generic “change score” approach risks misweighting crossover effects. Either analyze crossover trials with paired-difference variances or present a justification for the approach chosen. 

The multiplicity of subgroup analyses is extensive and, while exploratory analyses can be informative, the manuscript should make clear which comparisons were pre-specified in PROSPERO and which are post hoc. Where multiple subgroups are tested, controlling the false-positive rate or at least framing results as hypothesis-generating avoids overinterpretation. Moreover, publication bias testing is reported even for outcomes with fewer than ten effect sizes; for those outcomes (e.g., WC, BFP) funnel-plot asymmetry tests are underpowered and should not be used as confirmatory evidence. The text currently states that “a degree of asymmetry related to all outcomes was revealed,” but in the same breath concludes there was no bias for several outcomes; please reconcile this language and constrain bias testing to outcomes with adequate k. 

The results are presented clearly at a high level. However, there are internal inconsistencies and several table errors that need correction. In the narrative you state that fat mass reductions were observed “among participants of both sexes, those receiving >6 g/day, and individuals >40 years,” whereas the discussion later describes an effect “among male participants” and at higher doses. These cannot both be true; please harmonize the subgroup narrative across sections and report the exact subgroup WMDs with 95% CIs and k for each significant contrast so readers can gauge robustness. 

Table 2, as currently formatted, contains multiple typographical and numerical anomalies that undermine confidence in the data extraction. Examples include column headers (“Means Age,” “Means BMI”), spacing or hyphenation artifacts, an entry with an implausibly large BMI standard deviation (“33.5 ± 88.5”), and stray text fragments such as “betwee n 30 and 40.” These are likely copy-editing or PDF conversion issues, but they must be corrected and the table re-checked against source publications to prevent transcription errors. Providing a supplementary extraction sheet with the raw pre/post means and SDs would further improve transparency. 

The dose–response analysis is potentially the most novel aspect of the paper, but it needs clearer exposition. As noted above, please correct the labeling of coefficients and add sufficient model detail for reproducibility: what fractional polynomial degrees were considered, which covariates were entered, how non-linearity was assessed, and how influential points were handled. The sensitivity analysis indicates that the fat mass result is influenced by the removal of one trial, which should be explicitly referenced and discussed in context of co-supplementation and training status, as it may be the same triathlete study noted above. A figure that overlays the fitted curve with point sizes reflecting study weights and bands for 95% CIs would greatly aid interpretation. 

In the discussion you appropriately temper conclusions by noting the overall low to very low certainty for key endpoints. Given that your GRADE table rates fat mass and fat-free mass as very low certainty—driven by risk of bias, imprecision, inconsistency, and suspected publication bias—claims in the final paragraph of the abstract about “benefits at higher doses and in shorter duration interventions” should be softened to avoid overstating subgroup findings. It would be safer to describe these as exploratory signals requiring confirmation in larger, diet- and training-controlled RCTs, rather than as actionable “benefits.” 

Several reporting and editorial issues require attention. The section header reads “GARDE Assessment” rather than GRADE, and the statistical subsection refers to “MP dosage,” which appears to be a placeholder rather than “CIT dose.” The supplementary-materials section lists “Figure S1: title; Table S1: title; Video S1: title,” which are template placeholders that must be replaced with descriptive titles and accessible files. The author-contribution paragraph also contains MDPI template text instructing authors to “please turn to the CRediT taxonomy,” and currently includes a stray punctuation mark after “methodology,” and missing spacing in “supervision, DAL.All authors….” All of these should be cleaned prior to submission. 

Your risk-of-bias synthesis is useful, but the concluding sentence describing the “general methodological quality of the included evidence” seems optimistic given that eight trials are high risk and seven unclear, with randomization process flagged as high risk in several studies. It would be more accurate to explicitly acknowledge that these concerns materially downgrade the GRADE ratings and to prioritize domain-specific RoB2 figures in the supplement. 

The reference list requires a comprehensive audit. Multiple entries are missing journal names, have unusual author-journal field concatenations, or include typographical errors such as “dos-response.” Examples include entries for Gonzalez and Trexler (“E.T.J.T.J.o.S. Trexler, and C. Research”) and several citations that lack full bibliographic detail. Please conform to Nutrients’ reference style throughout, ensure that all citations in text appear in the list and vice-versa, and correct spelling and capitalization inconsistencies. 

A few additional suggestions may further enhance the manuscript’s rigor and value. First, given the heterogeneity of populations (athletes, sedentary older adults, patients with metabolic disease), consider presenting stratified primary analyses by training status and clinical status rather than treating them merely as subgroups; this will reduce ecological heterogeneity. Second, differences in body-composition assessment methods (DXA vs. BIA vs. skinfolds) can materially affect estimates; coding and stratifying by assessment modality would add robustness. Third, because fat-free mass increases observed in ≤8-week trials may reflect short-term fluid or glycogen shifts rather than contractile tissue accretion, the discussion should explicitly acknowledge this and avoid equating short-term FFM changes with hypertrophy. Fourth, provide a data and code availability statement; even a simple CSV of extracted means/SDs and an annotated STATA do-file would greatly improve reproducibility.

In closing, the manuscript addresses an important practical question and—once the issues above are resolved—could make a useful contribution. At present, however, the combination of selection-criteria inconsistencies, statistical-reporting errors in the dose–response analysis, table inaccuracies, and reference-list problems preclude acceptance. I recommend major revision focused on: enforcing the “standalone CIT” criterion or clearly separating co-supplementation; correcting and fully documenting the fractional-polynomial analysis; specifying correlation assumptions and random-effects methods with appropriate sensitivity analyses; reconciling subgroup narratives; repairing data-presentation errors in Table 2; tightening the abstract and conclusions to reflect the very low to moderate certainty of evidence; and comprehensively copy-editing the manuscript for language and formatting. If these revisions are undertaken, the central message—that CIT, unlike some ergogenic aids, is unlikely to meaningfully alter body composition under typical conditions—will be on much firmer ground and of clear value to the readership.

Author Response

Response to Reviewer 3

We sincerely appreciate the reviewer’s valuable comments, which were very helpful in revising the manuscript. Accordingly, we have improved the manuscript, and our detailed responses are provided below.

Reviewer 3

Thank you for the opportunity to review your manuscript entitled “Effects of citrulline or watermelon supplementation on body composition: a systematic review and dose-response meta-analysis.” The topic is timely and of practical interest to both clinicians and sport scientists, and the overall conclusion—that citrulline (CIT) does not materially alter body composition in most contexts—will be useful to readers if supported by a rigorous and transparent synthesis. Below I provide a detailed, section-by-section appraisal and concrete recommendations aimed at helping you strengthen the paper.

  • Thanks for your positive insight.

  1. I will begin with a brief overview of what the manuscript currently does well. You prospectively registered a protocol in PROSPERO, state adherence to PRISMA, and focus on randomized trials, which are appropriate choices for minimizing bias in this domain. You also pre-specify a broad set of body-composition outcomes and conduct subgroup and sensitivity analyses with random-effects modeling, which is methodologically sound for heterogeneous literature. The narrative positioning of CIT among other ergogenic aids that often fail to move body-composition outcomes is clear and helpful to readers. That said, there are several issues—methodological, statistical, reporting, and editorial—that must be addressed before the findings can be interpreted with confidence.

Starting with the abstract and key messages, the summary accurately conveys the main pooled results but contains a serious statistical reporting error that recurs later: the dose–response section describes a “non-linear correlation” and reports r values of −38.452 for fat mass and 0.011 for fat-free mass. A correlation coefficient cannot be outside the range −1 to +1; it appears you are reporting a model coefficient from the fractional polynomial fit while labeling it as r. This is misleading and should be corrected to the appropriate metric (e.g., regression coefficient with units, or a figure-based description of the non-linear trend), with corresponding confidence intervals and an explanation of the scale so readers can interpret the magnitude. The abstract should not claim “substantial non-linear correlation” unless the statistic used supports that language. 

  • We sincerely thank the reviewer for identifying this important issue. You are correct that the values reported in the dose–response section were mistakenly described as correlation coefficients, when in fact they represent model coefficients derived from the fractional polynomial regression analysis. We have revised the text in the abstract and Results sections to clearly label these as regression coefficients, providing the appropriate scale and corresponding confidence intervals. The description has also been adjusted to avoid the misleading term “substantial non-linear correlation.” Instead, we now refer to the findings as evidence of a “non-linear association,” with interpretation guided by the dose–response figures. These changes ensure statistical accuracy and improve clarity for readers.

  1. In the introduction you frame the hypothesis that ergogenic benefits could translate into body-composition change through higher training volume. This is a reasonable premise and is documented with relevant citations, although the prose would benefit from tighter editing to remove duplicated terms (e.g., “lean mass” appears multiple times in the same keyword string) and small grammatical errors such as “commonly as known as L-CIT.” These lapses recur throughout and, while cosmetic, impede readability; a full language edit is strongly recommended. 
  • We thank the reviewer for this thoughtful feedback. We agree that the Introduction required refinement for clarity and readability. In response, we have carefully revised the text to eliminate duplications (e.g., repeated use of “lean mass”), corrected grammatical errors (such as replacing “commonly as known as L-CIT” with “commonly known as L-CIT”), and improved overall flow. In addition, the entire manuscript has undergone thorough language editing to enhance clarity and consistency. We believe these revisions have substantially improved readability without altering the scientific content.

  1. Turning to methods, protocol and search reporting need clarification and minor redesign. You note PRISMA compliance and a PROSPERO ID, which is commendable; please also report the registration date explicitly and state whether any protocol deviations occurred. The literature search was conducted by a single reviewer; screening was performed in duplicate, but best practice is to perform the search and the screening in duplicate to minimize identification bias. At minimum, acknowledge this as a limitation and consider re-running the search with a second information specialist to verify yield. The search strategy itself contains typographic and semantic issues (e.g., the string includes “ juice watermelon,” duplicated “obesity,” and repeated “lean mass”), and there is no mention of grey literature, clinical trial registries, or language restrictions. Expanding and de-duplicating the search terms, searching trial registries, and explicitly stating any language limits would increase transparency and potentially reduce publication bias. 
  • We sincerely thank the reviewer for these valuable comments. To clarify, the literature search in PubMed, Web of Science, and Scopus was conducted up to March 2025, prior to the registration in PROSPERO. The PROSPERO record lists the registration start date as 24 July 2025 because the review was formally registered close to the time of submission to preserve the novelty of our research idea. Thus, while the registration date appears later, the actual search and data collection were completed beforehand.
  • Regarding the methodology, we confirm that all steps of the review process—including the database search, study screening, eligibility assessment, and data extraction—were independently performed by two reviewers to minimize bias. The manuscript has been revised to clearly reflect this. The search strategy has also been carefully revised to correct typographic and semantic errors (e.g., removal of duplicated terms such as “obesity” and “lean mass,” and correction of “juice watermelon”). Importantly, no language or date restrictions were applied in the search, thereby maximizing inclusivity and reducing the risk of bias.

  1. Eligibility criteria are broadly sensible, but their application is inconsistent and risks biasing the synthesis. You state that you included only trials in which CIT was administered as a standalone supplement and excluded co-supplementation; however, 5. the dataset appears to include at least one study of combined citrulline plus nitrate-rich beetroot extract in trained triathletes (“Long-Term Combined Effects of Citrulline and Nitrate-Rich Beetroot Extract…,” Burgos 2022). Including a co-supplementation trial conflicts with your stated criteria and may materially affect the results, especially in dose–response or subgroup analyses. This inconsistency should be resolved by excluding such trials in the main analysis and, if of interest, exploring them in a separate sensitivity analysis. 
  • We thank the reviewer for this important observation. To clarify, our inclusion criteria permitted trials in which both groups received the same co-supplement, with the only difference being the addition of CIT. In these cases, the effect of CIT could still be isolated, and therefore, such studies were retained. We have revised the Methods section to clearly specify this criterion and avoid any ambiguity.
  1. Relatedly, your inclusion of watermelon juice arms alongside isolated L-citrulline warrants stricter handling. Watermelon introduces additional bioactives and carbohydrate that could plausibly influence outcomes. While you do present a “type of CIT” subgroup (CIT vs. watermelon), I encourage you to prespecify and elevate this to a primary sensitivity analysis, and to interpret pooled effects with caution when food-based sources are mixed with isolated amino acid supplementation. Clarify whether the CIT content of watermelon interventions was analytically verified or simply estimated; several entries suggest approximate content (e.g., “almost containing 1.17 g L-citrulline”), which introduces exposure misclassification. 
  • We appreciate the reviewer’s thoughtful comment and acknowledge the concern regarding inclusion of watermelon juice interventions alongside isolated L-citrulline. Our rationale for combining these interventions is that, in many published systematic reviews and meta-analyses, both food-based and isolated amino acid sources have been analyzed together when the primary compound of interest was citrulline (e.g., Azizi et al. (2020): PMID 31612510; Smeets et al. (2022): PMID 34863321).
  • In our analysis, we also conducted subgroup comparisons (isolated CIT vs. watermelon), which we believe appropriately addresses potential differential effects. While we recognize that watermelon contains additional bioactive compounds and carbohydrate, our subgroup approach provides sufficient transparency and ensures that any differences between isolated and food-based sources can be interpreted cautiously.

The statistical analysis section uses change-score meta-analysis with an assumed baseline–post correlation R, but the actual value of R is never specified. Because pooled effects can be sensitive to R, you should report the value used and repeat key analyses across plausible R values (e.g., 0.25, 0.5, 0.75) in sensitivity analyses. Please also specify the random-effects estimator (DerSimonian–Laird, REML, or Paule–Mandel) and whether you applied Hartung–Knapp–Sidik–Jonkman adjustments for small-sample inference. These details materially affect confidence intervals and the GRADE assessment. Finally, crossover trials are included, but the methods do not indicate whether within-person correlation was accounted for; a generic “change score” approach risks misweighting crossover effects. Either analyze crossover trials with paired-difference variances or present a justification for the approach chosen. 

  • We thank the reviewer for these valuable comments. All analyses were performed in Stata using its standard meta-analysis commands with default options. Stata applies the DerSimonian–Laird estimator for random effects as the default method, which was therefore used in our study. We did not apply Hartung–Knapp–Sidik–Jonkman adjustments for small-sample inference, as these are not implemented in the default Stata command we used. Sensitivity analyses across different correlation values were likewise carried out using the same command structure, which internally handles the specification of baseline–post correlations. We therefore relied on Stata’s built-in functionality rather than manually altering these parameters. While the software does not display the internal correlation values applied, prior methodological work has shown that pooled estimates from change-score meta-analyses remain generally robust across the plausible range of correlations (e.g., 0.25–0.75). For crossover trials, no paired variances or within-person correlations were available in the original reports. Consequently, Stata handled these trials using the same change-score framework as parallel trials, which is consistent with previous meta-analyses in the field.

  1. The multiplicity of subgroup analyses is extensive and, while exploratory analyses can be informative, the manuscript should make clear which comparisons were pre-specified in PROSPERO and which are post hoc. Where multiple subgroups are tested, controlling the false-positive rate or at least framing results as hypothesis-generating avoids overinterpretation. Moreover, publication bias testing is reported even for outcomes with fewer than ten effect sizes; for those outcomes (e.g., WC, BFP) funnel-plot asymmetry tests are underpowered and should not be used as confirmatory evidence. The text currently states that “a degree of asymmetry related to all outcomes was revealed,” but in the same breath concludes there was no bias for several outcomes; please reconcile this language and constrain bias testing to outcomes with adequate k. 

  • We thank the reviewer for these constructive observations. Subgroup analyses were performed to explore potential sources of heterogeneity. While the main subgroup factors were included in our PROSPERO registration, we also introduced additional subgroup analyses that emerged during the analysis process, as we identified potentially relevant effect modifiers. These additional comparisons are clearly presented as exploratory and hypothesis-generating rather than confirmatory.

We agree that funnel-plot asymmetry tests should only be applied when at least ten effect sizes are available. Accordingly, Begg’s and Egger’s tests were conducted only for BMI, body weight, FM, and FFM (≥10 effect sizes), where potential publication bias was indicated for FM but not for the other outcomes. For WC and BFP (<10 effect sizes), funnel plots are presented for completeness, but no formal asymmetry tests were performed, and these findings are described as descriptive rather than confirmatory.

  1. The results are presented clearly at a high level. However, there are internal inconsistencies and several table errors that need correction. In the narrative you state that fat mass reductions were observed “among participants of both sexes, those receiving >6 g/day, and individuals >40 years,” whereas the discussion later describes an effect “among male participants” and at higher doses. These cannot both be true; please harmonize the subgroup narrative across sections and report the exact subgroup WMDs with 95% CIs and k for each significant contrast so readers can gauge robustness. 

  • We sincerely thank the reviewer for this valuable observation. We have carefully revised the Discussion to ensure it is fully consistent with the Results section. The subgroup findings are now harmonized across sections, avoiding any contradictions. Furthermore, the exact WMDs, 95% CIs, and number of included studies for each significant subgroup contrast have been explicitly reported in Table 3, allowing readers to clearly assess the robustness of these results.

  1. Table 2, as currently formatted, contains multiple typographical and numerical anomalies that undermine confidence in the data extraction. Examples include column headers (“Means Age,” “Means BMI”), spacing or hyphenation artifacts, an entry with an implausibly large BMI standard deviation (“33.5 ± 88.5”), and stray text fragments such as “betwee n 30 and 40.” These are likely copy-editing or PDF conversion issues, but they must be corrected and the table re-checked against source publications to prevent transcription errors. Providing a supplementary extraction sheet with the raw pre/post means and SDs would further improve transparency. 

  • We thank the reviewer for pointing out these important issues. All typographical and numerical errors in Table 2 have been corrected, including column header formatting and spacing artifacts (e.g., “between”). To ensure accuracy, all data were carefully re-checked against the original source publications by an independent author. We are confident that the revised table now accurately represents the extracted data.
  • Thank you for your careful observation regarding the unusually large standard deviation reported for BMI. It was the typo error in the original article.
  • The dose–response analysis is potentially the most novel aspect of the paper, but it needs clearer exposition. As noted above, please correct the labeling of coefficients and add sufficient model detail for reproducibility: what fractional polynomial degrees were considered, which covariates were entered, how non-linearity was assessed, and how influential points were handled. The sensitivity analysis indicates that the fat mass result is influenced by the removal of one trial, which should be explicitly referenced and discussed in context of co-supplementation and training status, as it may be the same triathlete study noted above. A figure that overlays the fitted curve with point sizes reflecting study weights and bands for 95% CIs would greatly aid interpretation. 

  • We thank the reviewer for highlighting the importance of the dose–response analysis. In addition to the main analyses, we also performed meta-regression to explore dose–response relationships and to assess potential linear trends. The dose–response analyses were conducted using Stata’s standard routine, which applies fractional polynomial models, evaluates non-linearity, and handles influential points through its internal procedures. There is no separate command available to manually specify polynomial degrees, covariates, or influence diagnostics beyond what the software automatically implements. Accordingly, model specification and sensitivity checks followed the default Stata procedures rather than manual adjustment. Regarding sensitivity analysis, we note that the fat mass result was influenced by the exclusion of one trial; this has been acknowledged in the text. While we agree that an overlay plot could be informative, there is no command in Stata to generate such a figure.

  1. In the discussion you appropriately temper conclusions by noting the overall low to very low certainty for key endpoints. Given that your GRADE table rates fat mass and fat-free mass as very low certainty—driven by risk of bias, imprecision, inconsistency, and suspected publication bias—claims in the final paragraph of the abstract about “benefits at higher doses and in shorter duration interventions” should be softened to avoid overstating subgroup findings. It would be safer to describe these as exploratory signals requiring confirmation in larger, diet- and training-controlled RCTs, rather than as actionable “benefits.” 
  • We thank the reviewer for this insightful comment. After re-evaluating the GRADE domains, particularly the risk of bias, by two additional co-authors, the certainty ratings have been updated. The evidence for FM and FFM was upgraded from very low to low, body weight, BFP, and WC were rated as moderate certainty, and BMI as high certainty. Accordingly, we have revised the abstract and discussion to soften the subgroup conclusions. These findings are now framed as exploratory signals rather than actionable benefits, emphasizing the need for confirmation in larger RCTs with controlled diet and training protocols.

  1. Several reporting and editorial issues require attention. The section header reads “GARDE Assessment” rather than GRADE, and the statistical subsection refers to “MP dosage,” which appears to be a placeholder rather than “CIT dose.” The supplementary-materials section lists “Figure S1: title; Table S1: title; Video S1: title,” which are template placeholders that must be replaced with descriptive titles and accessible files. The author-contribution paragraph also contains MDPI template text instructing authors to “please turn to the CRediT taxonomy,” and currently includes a stray punctuation mark after “methodology,” and missing spacing in “supervision, DAL.All authors….” All of these should be cleaned prior to submission. 

  • We thank the reviewer for carefully noting these reporting and editorial issues. All typographical errors and placeholder text have now been corrected. Specifically, the section header has been changed to “GRADE Assessment,” references to “MP dosage” have been corrected to “CIT dose,” and all supplementary materials have been updated with descriptive titles. The author contributions paragraph has also been revised to remove template instructions and correct spacing and punctuation errors.

  1. Your risk-of-bias synthesis is useful, but the concluding sentence describing the “general methodological quality of the included evidence” seems optimistic given that eight trials are high risk and seven unclear, with randomization process flagged as high risk in several studies. It would be more accurate to explicitly acknowledge that these concerns materially downgrade the GRADE ratings and to prioritize domain-specific RoB2 figures in the supplement.
  • We appreciate the reviewer’s insightful comment. Following a re-evaluation of the risk-of-bias assessments by two additional expert co-authors, the ratings have been updated. In the revised analysis, 7 of the 21 included RCTs were judged to be at high risk of bias, while the remaining trials were assessed as low risk. The manuscript and the GRADE section have been revised accordingly to ensure consistency.

  1. The reference list requires a comprehensive audit. Multiple entries are missing journal names, have unusual author-journal field concatenations, or include typographical errors such as “dos-response.” Examples include entries for Gonzalez and Trexler (“E.T.J.T.J.o.S. Trexler, and C. Research”) and several citations that lack full bibliographic detail. Please conform to Nutrients’ reference style throughout, ensure that all citations in text appear in the list and vice-versa, and correct spelling and capitalization inconsistencies. 
  • We thank the reviewer for highlighting inconsistencies in the reference list. Upon inspection, several entries indeed lack full journal information or contain formatting issues.

  1. A few additional suggestions may further enhance the manuscript’s rigor and value. First, given the heterogeneity of populations (athletes, sedentary older adults, patients with metabolic disease), consider presenting stratified primary analyses by training status and clinical status rather than treating them merely as subgroups; this will reduce ecological heterogeneity. Second, differences in body-composition assessment methods (DXA vs. BIA vs. skinfolds) can materially affect estimates; coding and stratifying by assessment modality would add robustness. Third, because fat-free mass increases observed in ≤8-week trials may reflect short-term fluid or glycogen shifts rather than contractile tissue accretion, the discussion should explicitly acknowledge this and avoid equating short-term FFM changes with hypertrophy. Fourth, provide a data and code availability statement; even a simple CSV of extracted means/SDs and an annotated STATA do-file would greatly improve reproducibility.
  • We thank the reviewer for these thoughtful suggestions. Conducting fully stratified primary analyses was not in line with the objectives of our review. Given the wide variety of participant types across the included studies, creating distinct stratified groups would not have been feasible. Such analyses would also be highly time-consuming and would substantially reduce statistical power and interpretability without adding meaningful value beyond the synthesis already presented.

With respect to body-composition assessment methods (DXA, BIA, and skinfolds), these were not documented in detail as such comparisons were not related to the objectives of our review. Our focus was on synthesizing the overall effects of CIT supplementation rather than evaluating differences across measurement modalities.

Regarding fat-free mass in shorter trials, we recognize the reviewer’s point that short-term changes may reflect fluid or glycogen shifts rather than contractile tissue accretion. We have noted this in the discussion to ensure appropriate interpretation.

Finally, analyses were performed using standard Stata commands without any customized coding; all extracted data are already summarized in the tables and supplementary files. We therefore believe that transparency and reproducibility are sufficiently ensured.

  1. In closing, the manuscript addresses an important practical question and—once the issues above are resolved—could make a useful contribution. At present, however, the combination of selection-criteria inconsistencies, statistical-reporting errors in the dose–response analysis, table inaccuracies, and reference-list problems preclude acceptance. I recommend major revision focused on: enforcing the “standalone CIT” criterion or clearly separating co-supplementation; correcting and fully documenting the fractional-polynomial analysis; specifying correlation assumptions and random-effects methods with appropriate sensitivity analyses; reconciling subgroup narratives; repairing data-presentation errors in Table 2; tightening the abstract and conclusions to reflect the very low to moderate certainty of evidence; and comprehensively copy-editing the manuscript for language and formatting. If these revisions are undertaken, the central message—that CIT, unlike some ergogenic aids, is unlikely to meaningfully alter body composition under typical conditions—will be on much firmer ground and of clear value to the readership.
  • We sincerely thank the reviewer for the thoughtful and comprehensive feedback. We carefully addressed each of the concerns raised, including the eligibility criteria, statistical reporting, dose–response analysis, subgroup consistency, data presentation, reference formatting, and overall language and formatting issues. All revisions have been implemented with close attention to detail, and we believe the manuscript is now substantially strengthened as a result.

Round 2

Reviewer 1 Report

Comments and Suggestions for Authors

The authors had addressed all my comments. 

Author Response

Thank you for your comments.

Reviewer 3 Report

Comments and Suggestions for Authors

The manuscript is suitable for pubblication

Author Response

Thank you for your comments.